# Analysis on the structure effect of marine fishery total factor productivity under high-quality development in China

**Bo Wang[1], Limin Han[2], Hongzhi Zhang[3] ***

**1** School of Economic and Management, Yantai University, Yantai, Shandong, China, **2** Management College, Ocean University of China, Qingdao, Shandong, China, **3** Shandong Foreign Trade Vocational College, Qingdao, Shandong, China

\* ouqdsnow@163.com

**Data Availability Statement:** All relevant data are within the manuscript and its Supporting Information files.

## Abstract

Improving total factor productivity (TFP) is the source of power for high-quality development. Industrial structure optimization is an important way to improve TFP. This paper constructed an econometric model of industry structure changes impacting on TFP in the marine fisheries and conducted an empirical test and analysis. The results showed that the industry rationalization, softening and processing coefficient of marine fishery had a significant "structural dividend" for improving its TFP; while the impact of industrial structure advancement and aquaculture-catching structure changes did not have "structural dividend", but it could be a combination of other factors to reduce these adverse effects. We believe that simply pursuing the advanced evolution of the industrial structure is not conducive to sustainable development of fishery. Under the pursuit of the rationalization of the marine fishery industry structure, by promoting the coordinated evolution of marine fisheries advancement, aquaculture-catching structure and other factors, the "structural dividend" effect can be enhanced and the fishery can achieve sustainable development. Finally, it proposed to promote the development of advancement and rationalization of marine fishery industry structure coordinately, adjust fishery science and technology transformation direction and key points, and accelerate the development of intensive processing industry by cross-border integration.

## Introduction

The economy of China has changed from high-speed growth stage to high-quality development stage, and high-quality development has become a long-term strategy followed by the national modernization construction in the new era [1]. Ocean is a strategic place for high-quality development, and marine economy plays an important role in economic reform. As one of the important industries of marine economy, marine fishery plays a significant role in ensuring food supply, optimizing dietary structure and promoting the construction of a maritime power. Over the past 40 years of reform and development, the comprehensive capacity and food supply capacity of marine fisheries have been significantly enhanced, especially in the

**Funding:** Youth Project of Natural Science Foundation of Guangxi Science and Technology Department (Number: 2018GXNSFBA050010) and Youth Project of National Social Science Foundation of China (Number: 19CJY023) provide support in study design and analysis. The paper is one of the phased research results of this project (Youth Project of Natural Science Foundation of Guangxi Science and Technology Department (Number: 2018GXNSFBA050010). Guangxi Key R & D Plan (Number: Guike AB1850023)provided basic materials for writing the thesis (such as marine fishery development data). Yantai University Ph.D. Started Funding Project (Number: 1103-2220004JG20B79) had a role in study design, decision to publish, or preparation of the manuscript.

**Competing interests:** The authors have declared that no competing interests exist.

total amount of marine products increased from 3.59 million tons in 1978 to 33.01 million tons in 2018 [2]. However, in the process of fishery development, it is also faced with the problems of resource decline, water pollution aggravation, lack of kinetic energy and so on. The report of the 19th National Congress of the Communist Party of China clearly pointed out that "with the supply side structural reform as the main line, we should promote the quality change, efficiency change and dynamic change of economic development, and improve the total factor productivity". Improving the total factor productivity (TFP) has become the power source of promoting high-quality economic development. At the same time, the Ministry of Agriculture and Rural Affairs of the People's Republic of China has carried out a series of deployment on the high-quality development of fisheries, accelerating the transfer of fisheries from scale expansion to quality benefit. In 2020, the Ministry of Agriculture and Rural Affairs of the People's Republic of China issued the "key points of fishery and fishery administration in 2020", which pointed out that "persistently stabilizing quantity, improving quality, changing mode and protecting ecology". Enhancing TFP is an important way to improve fishery quality. It can fundamentally change the mode of production, improve labor efficiency, and provide new impetus for the transformation and upgrading of marine fisheries. With the supply-side structural reforms in fishery, the structural adjustment of fishery industry will directly or indirectly affect the TFP of marine fishery by accelerating the flow of factors, changing the technical efficiency and influencing the degree of industrial division [3]. With the reform of the economic system, the development of marine fishery should not only focus on the problem of growth, but also pay more attention to the quality of development. Improving TFP has become one of the important demands for the high-quality development of marine fishery economy.

TFP is an important indicator to measure the quality of economic development [4], and it is a hot field in academic research. TFP refers to the increase in output after deducting the growth brought by labor, capital, land and other factors. It is the comprehensive ability of industrial technology progress, management innovation, product quality improvement and structure advancement [5]. At present, the methods to measure TFP mainly include Solow residual method [6–8], stochastic frontier production function model [9–11], data envelopment analysis method [12–15], DEA-Malmquist index after optimization [16,17], and the four-component stochastic boundary model [18]. The transformation and upgrade of industrial structure is an important manifestation of economic growth [19], and promotes the transfer of factors such as capital, and labor from the low-productivity sector to the high-productivity sector, thereby improving the production efficiency of the entire sector [20]. Therefore, it is also an important factor affecting TFP. Some scholars have analyzed and affirmed the impact of industrial structure adjustment on economic production efficiency, but there are significant differences in research conclusions. Most studies indicate that the advancement of industrial structure has a positive effect on the improvement of production efficiency, which confirms the "structural dividend hypothesis" [21,22], but this positive relationship will be different due to regional economic development differences [23,24]. Lu et al. thought that the rationalization and optimization of industrial structure has a positive effect on promoting the development of green TFP in China and all the regions, but the regional characteristics of this effect are different. The effect of rationalization of industrial structure decreases from the western region to the central and eastern regions, but the effect of optimization of industrial structure are increases from the western region to the central and eastern regions [25]. Li and Lin believed that the structural changes in the manufacturing industry have positive effects on energy adjusted TFP [26]. However, some studies indicate that the effect of industrial structure change on productivity is not significant, and there is no "structural dividend hypothesis" [27,28]. From the perspective of micro factors, some scholars have confirmed that there is no "structural dividend hypothesis" in the transfer of production factors such as labor force [29] and capital [30]. Zhang et al. studied the contribution of labor and capital input to the

growth of China's total factor productivity. The results showed that the growth of capital productivity exhibited a remarkable slowdown after the mid-1990s; although labor productivity continued to increase, the relative labor efficiency between provinces has declined after 2000 [31].

The research on fishery TFP mainly focuses on the measurement, evaluation and influencing factors of TFP. Under the influence of sample selection, there are differences in the research results of fishery TFP. The fishery TFP generally shows an increase [32] or decrease [33] trend in a specific period. From the analysis of contribution to fishery TFP, some studies indicate that the improvement of TFP benefits from the joint effect of technical efficiency change and technological progress [34], but others indicate that the improvement of TFP comes from the unilateral contribution of technical efficiency [35] or technical progress [36]. Some studies also analyze other factors such as the management system [37] and resource allocation [38] on the influence of fishery FP. At the same time, some scholars analyzed the convergence of marine fishery TFP. But there is little research on the impact of industrial structure changes on the TFP of marine fishery. Ji thought that the high proportion of the primary industry of marine fishery is an important factor for the continuous downturn of the TFP of marine fishery. The key to improve the TFP of marine fishery is to develop the secondary and tertiary industries of marine fishery [39]. Zheng et al. used DEA-Malmquist index to analyze fishery TFP in 12 coastal areas of mainland China and Taiwan. In general, changes in technical efficiency (TEC)and technological progress (TP) had promoted the increase in fishery TFP. However, TEC promoted and TP hindered the TFP of capture fisheries on both sides of the Taiwan Strait; in aquaculture, TE and TEC promoted TFP growth [40]. Solis et al. analyzed the relationship between the total factor productivity of Mexico red snapper commercial fishery and individual fishing quota (IFQ), and believed that IFQ had a positive impact on the total factor productivity of fleet fishing and that most of the productivity gains were due to improvements in technical efficiency [41].

Generally speaking, there are few researches on TFP of marine fishery, which mainly focus on the measurement and evaluation of TFP. Few systematic analysis is carried out on the relationship between industrial structure adjustment and TFP. Therefore, based on the perspective of high-quality development and sustainable development of marine fisheries, we mainly analyze the structural effect of TFP of marine fishery from the perspectives of advanced, rationalized, softened and production structure of marine fishery industry, rationally judges whether structural adjustment produces "structural dividend" on TFP of marine fishery, and clarifies the relationship between industrial structure adjustment and TFP of marine fishery, seeks the power point in the process of fishery structure reform.

## Overview of marine fishery development in China

With the improvement of China's fishery technological innovation capabilities, marine fisheries operations capabilities have gradually increased, and significant achievements have been made in industrial development. China has become the world's largest seafood producer. The marine fisheries development has enhanced its economic, social and ecological functions, and the comprehensive benefits of it have been continuously improved. Firstly, its economic contribution and food supply capacity continue to increase. Secondly, the industrial structure is gradually optimized. Thirdly, the marginal efficiency of production factors is increasing, and seafood is gradually diversified.

### Enhancement in marine fisheries production and economic value

From an overall perspective, the economic strength and contribution of marine fisheries have continued to increase. In 2016, the total economic value of fishery in the whole society was

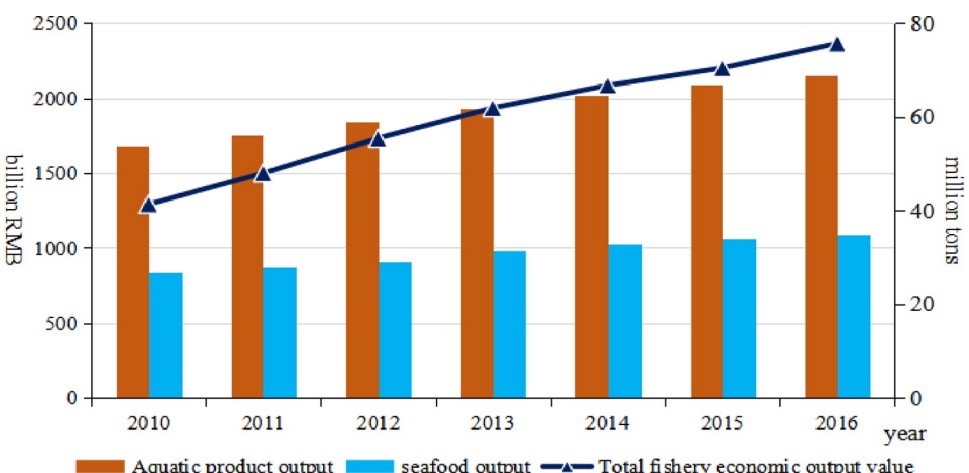

**Fig 1. The changes in the total output value of fishery economy and the output of aquatic products in China [2].**

2366.23 billion RMB (Fig 1), an increase of 83.01% over 2010, accounting for 37.16% of the total output value of agriculture, forestry, animal husbandry and fishery. The import and export trade of aquatic products is increasing day by day. In 2016, the fishery total import and export trade was 30.11 billion dollars, accounting for 0.82% of the total domestic import and export trade, an increase of 1 percentage point over 2015, and the import and export volume of aquatic products remains at about 800 tons. The food supply capacity of marine fisheries has increased significantly. In 2016, the total output of world marine aquatic products reached 102.8 million tons, accounting for 65.7% of the total global aquatic product output. China's marine aquatic products amounted to 34.90 million tons, accounting for 33.95% of the world's marine aquatic products. It can provide 2.66 million tons of animal protein, accounting for about 13% of all animal protein produced in China, and the amount of animal protein supply is increasing at an average annual rate (3.34%) higher than the average annual growth rate of terrestrial ecosystems (2.52%).

From regional perspective, China's coastal areas which the total economic value of fisheries accounted for 73.93% of the total economic value of the national fishery in 2016 become a pioneer area for fishery economic growth. However, there are significant differences in regional development. Among the areas with a total fishery economic output value of more than 200 billion RMB, coastal areas account for 80%, mainly including Shandong (390.22 billion RMB), Jiangsu (302.06 billion RMB), Guangdong (286.31 billion RMB), Fujian (273.41 billion RMB). Among them, the total fishery economic output value in Jiangsu and Shandong provinces exceeds 300 billion RMB, but Tianjin, Hebei, Guangxi, and Hainan are less than 100 billion RMB (Fig 2).

## Coordination of fishery industry structure

With the rapid development of the secondary and tertiary fishery industries, the proportion of the three types of industry structure has evolved from 52.22: 23.89: 23.89 in 2010 to 50.73: 22.87: 26.40 in 2016(Fig 3). The growth rate of the proportion of the tertiary fishery industry exceeds that of the primary and secondary industries, and the trend of fishery economy as a service is gradually obvious, but the primary fishery industry is still the leading industry.

From the internal analysis of the three major industrial systems, the primary fishery industry is aquaculture as the main industry. In 2016, the output value of mariculture was 314.39 billion RMB, an increase of 116.317 billion RMB over the output value of seawater fishing (Fig 4).

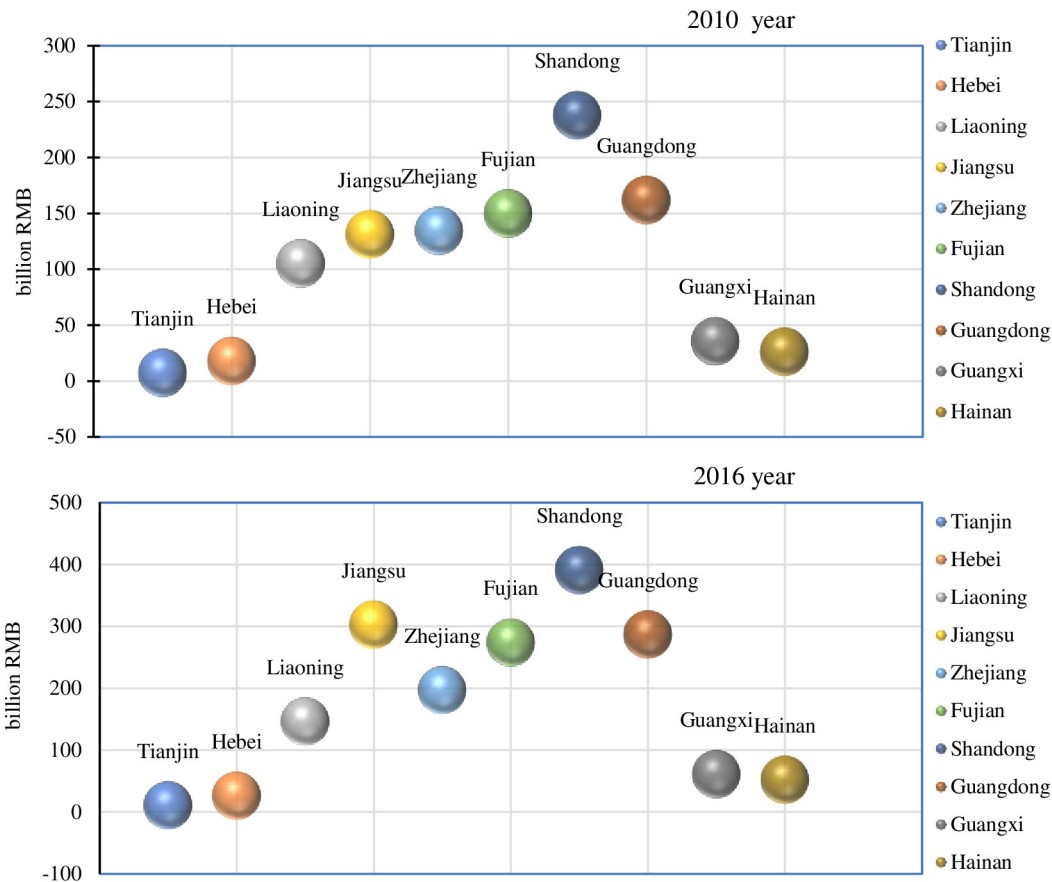

**Fig 2. The gross economic output of fishery in China's main coastal areas (2010, 2016).**

The secondary fishery industry is dominated by aquatic product processing, and other industries have also developed rapidly with the expansion of industrial scale. In 2016, the output value of aquatic product processing reached 409.023 billion RMB, accounting for 75.60% of the secondary fishery industry (Fig 5). The tertiary fishery industry focuses on the circulation

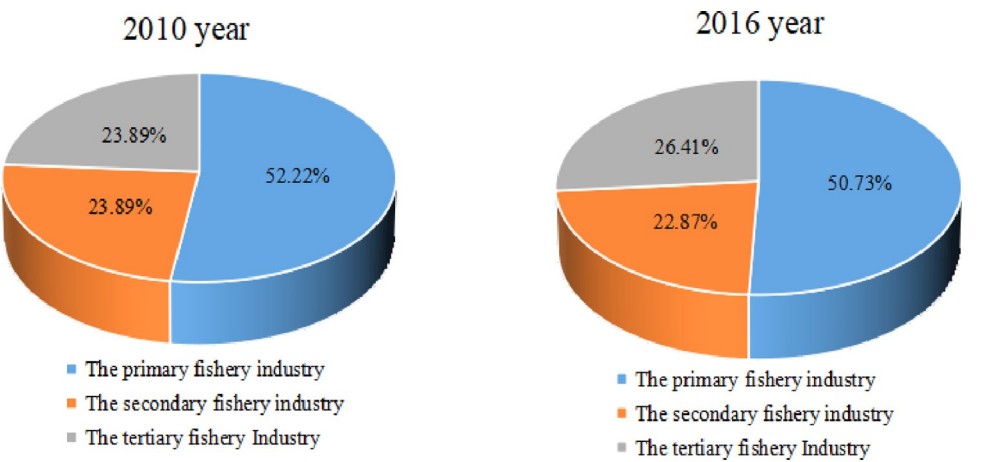

**Fig 3. The proportion of the three types of industry structure in China (2010, 2016).**

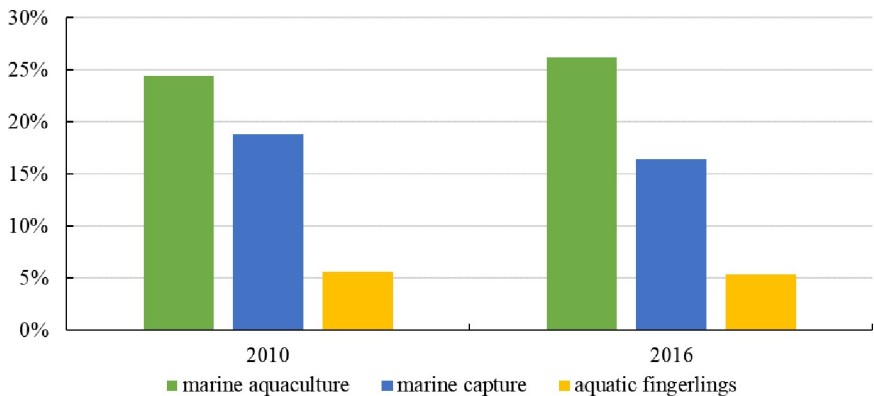

**Fig 4. Marine fishery industrial distribution in the primary fishery industry (2010, 2016).**

of aquatic products. From 2010 to 2016, the proportion of aquatic product circulation in the fishery tertiary industry output value has always remained between 80% and 82% (Fig 6), becoming the main sector that stimulates the fishery tertiary industry.

## Temporal and spatial characteristics of TFP of Marine Fisheries

Based on the CRS model and DEA-Malmquist index, we calculate the change data of total factor productivity (*tfp*), technical efficiency (*effch*), technological progress (*techch*) and scale efficiency (*sech*) of marine fisheries based on CRS model, and analyze the difference characteristics from two dimensions of time and space.

**Time variation characteristics.** According to the data calculation, the average annual total factor productivity of marine fishery is 1.005, which indicates that the total factor productivity has increased by 0.5% annually, showing an overall growth trend, but the growth rate is not large. Affected by the financial crisis in 2008, the total factor productivity of marine fishery declined greatly from 2007 to 2008, because the financial crisis led to market instability, sluggish industrial development and reduced resource factor utilization efficiency; during 2009–2016, the fluctuation range was relatively low and basically remained stable, which was in

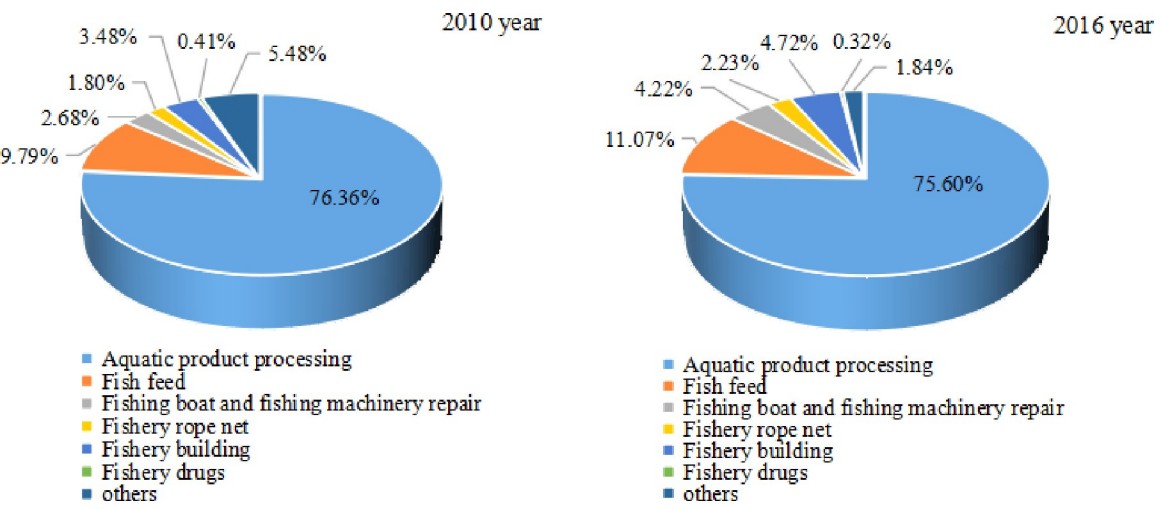

**Fig 5. Industrial distribution in the secondary fishery industry (2010, 2016).**

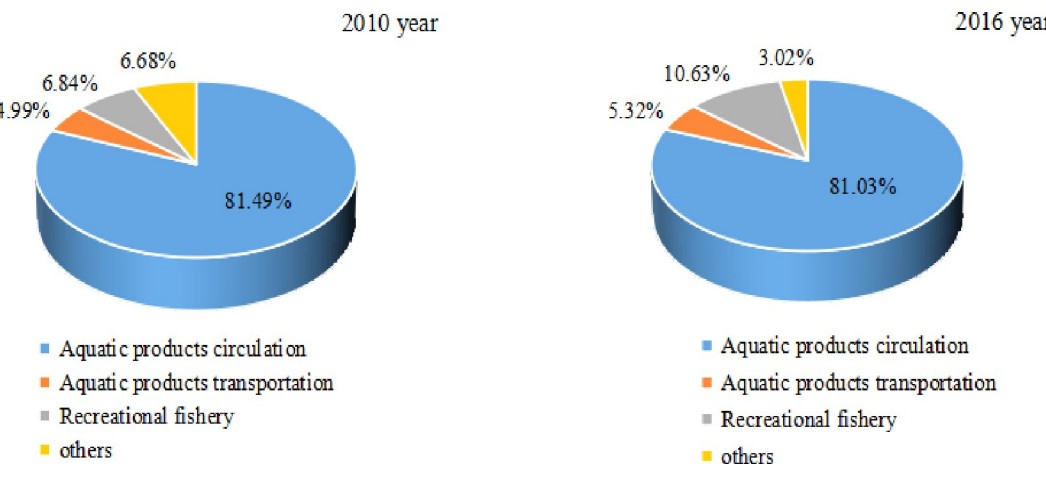

**Fig 6. Industrial distribution in the tertiary fishery industry (2010, 2016).**

contradiction with China's economy entering the new normal and supply side structural contradiction, and the economic development speed was loose. There is a great correlation between them. During the period of 2010–2011, 2012–2013 and 2014–2015, the TFP of marine fishery was lower than 1.000, and its decline was much lower than that of 2007–2008 (Fig 7).

From the decomposition results of TFP Index, we can see that the average changes of technical efficiency and technological progress of marine fishery are 1.006 and 0.995 respectively, and the change of technical efficiency of marine fishery is greater than 1.000, which indicates that the technical efficiency of China's marine fishery is improved, but the change of technological progress shows a downward trend from 2003 to 2016, indicating that the technological progress of marine fishery retrogressed comparing with the technical demand of industrial development. Therefore, it can be said that the improvement of TFP of Marine Fisheries in 2003–2016 benefited from the improvement of technical efficiency of Marine Fisheries rather than technological progress.

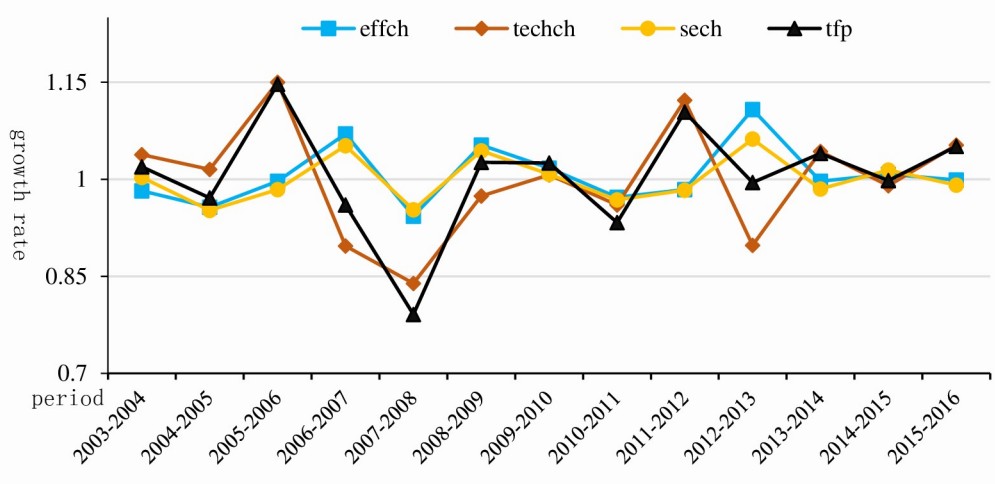

**Fig 7. Trend of total factor productivity of marine fishery in China from 2003 to 2016.**

**Table 1. Summary of the means of Malmquist index of marine fisheries in various coastal regions.**

| regions | effch | techch | sech | TFP | TFP Sequence | C.V. |
|---------|-------|--------|------|-----|--------------|------|
| Tianjin | 1.001 | 1.014 | 1.001 | 1.015 | 5 | 0.320 |
| Hebei | 0.983 | 0.985 | 0.983 | 0.969 | 10 | 0.169 |
| Liaoning | 1.000 | 0.975 | 1.000 | 0.975 | 9 | 0.150 |
| Jiangsu | 1.051 | 0.995 | 1.009 | 1.045 | 1 | 0.078 |
| Zhejiang | 1.000 | 1.016 | 1.000 | 1.016 | 4 | 0.192 |
| Fujian | 1.000 | 0.979 | 1.000 | 0.979 | 8 | 0.105 |
| Shandong | 1.000 | 0.99 | 1.000 | 0.990 | 6 | 0.062 |
| Guangdong | 1.024 | 0.993 | 1.001 | 1.017 | 3 | 0.072 |
| Guangxi | 1.000 | 0.985 | 1.000 | 0.985 | 7 | 0.075 |
| Hianan | 1.000 | 1.022 | 1.000 | 1.022 | 2 | 0.347 |

**Characteristics of spatial differences.** The difference of regional development level and marine fishery resource endowment will lead to significant spatial difference of total factor productivity of marine fishery. Table 1 shows the difference characteristics of the means of Malmquist index including total factor productivity, technical efficiency and technological progress of Marine Fisheries in coastal areas (except Shanghai). It can be seen from the previous part of this paper that the annual average TFP of Marine Fisheries in China from 2003 to 2016 was 1.005, and Jiangsu, Hainan, Guangdong, Zhejiang, Tianjin and other regions exceeded the national average level, with Jiangsu being the highest, reaching 1.045, indicating that the average annual total factor productivity of Marine Fisheries in these areas has shown an overall increase trend. However, the average TFP of Marine Fisheries in Shandong, Guangxi, Fujian, Liaoning, Hebei and other regions are all lower than 1.000, which indicates that the average annual TFP of Marine Fisheries in these areas shows a downward trend. At the same time, using coefficient of variation (C.V.) to analyze the change of total factor productivity of marine fishery in different regions from 2003 to 2016. The coefficient of variation of Hainan, Tianjin, Zhejiang, Hebei, Liaoning, Fujian and other regions are higher than those of Jiangsu, Guangxi, Guangdong, Shandong and other regions, which shows that the interannual variation of TFP of marine fishery in Hainan, Tianjin, Zhejiang, Hebei, Liaoning, Fujian is relatively high. The reason for the large fluctuation may be the upgrading of regional marine fishery technology and the adjustment of industrial structure.

From the decomposition results of total factor productivity, in terms of technical efficiency, the technical efficiency of Marine Fisheries in Tianjin, Jiangsu and Guangdong is greater than 1.000, which indicates that the technical efficiency of marine fishery has been gradually improved during 2003–2016, while that of Hebei, Liaoning, Zhejiang, Fujian, Shandong, Guangdong, Hainan and other regions is 1.000, which indicates that the technical efficiency of these areas has not changed in the period of 2003–2016. In terms of technological progress, the technological progress of Marine Fisheries in Tianjin, Zhejiang and Hainan has changed by more than 1.000, which indicates that the marine fisheries in these areas have expanded their frontier technologies through technological innovation, and expanded the production potential areas of marine fisheries; while the technological progress changes of Marine Fisheries in other coastal areas are less than 1.000, indicating that the marine fishery technology in these areas has retrogressed.

To sum up, the improvement of total factor productivity of marine fishery in Tianjin is the result of the improvement of technical efficiency and technological progress; the improvement of TFP of marine fishery in Jiangsu and Guangdong mainly depends on the improvement of technical efficiency, while the improvement of TFP of marine fishery in Zhejiang and Hainan is mainly driven by the improvement of technological progress.

## Experimental design

**Variable selection and setting.** The main variables involved in this paper include total factor productivity and industrial structure of marine fishery.

The explained variable is total factor productivity of marine fishery. Based on the CRS model, this paper uses DEA-Malmquist to calculate the total factor productivity of marine fishery. Before calculating Malmquist index, it is necessary to select economic input and output indicators reasonably and objectively. The input indicators of marine fishery mainly include labor force, capital, technology, fishing vessel, mariculture area, etc. Among them, marine fishery labor force is measured by the year-end number of marine fishery employees; for marine fishery capital, due to the lack of marine fishery capital input data in the relevant Yearbook, this paper uses a method that the fixed asset investment amount of agriculture, forestry, animal husbandry and fishery(AFAF) multiplied by the proportion of marine fishery in AFAF to measure the level of capital investment in marine fishery, and adopts the method of calculating capital stock by Wang et al. [42] for reference, calculates the capital stock of marine fishery and takes it as the input index. Fishing vessel and mariculture area have important support for the development of marine fishery economy, so they are taken into consideration, and the indicator data comes from "China Fishery Statistical Yearbook". Marine fishery science and technology is the key driving force to promote its high-quality development. However, due to the lack of relevant data on R & D investment of marine science and technology in relevant yearbooks, the funds for marine fishery technology promotion are used for indirect measurement.

The main explanatory variables are the industrial structure of marine fishery. In this paper, the industrial structure is advanced, rationalized, softened, aquaculture structure and processing coefficient are measured. The specific calculation methods are as follows:

The advancement of marine fishery industry structure (*sadvance*). It is measured by Moore structural variation coefficient adjusted [43]. Firstly, all industries are divided into three categories according to the standard of three industries, and the vector $X_0 = (x_{1,0} \ x_{2,0} \ x_{3,0})$ is constructed by taking the contribution of the three industries to GDP as the component; secondly, the angles $a_j$ ($j = 1,2,3$) between $X_0$ and the vectors arranged from low level to high level are calculated respectively $X_1 = (1 \ 0 \ 0)^T$, $X_2 = (0 \ 1 \ 0)^T$; $X_3 = (0 \ 0 \ 1)^T$. The calculation formula is as follows:

$$\alpha_j = arccos\left( \sum_{i=1}^{3}(x_{i,j} \times x_{i,0}) / \sqrt{\sum_{i=1}^{3}x_{i,j}^2 \times \sum_{i=1}^{3}x_{i,0}^2} \right) \tag{1}$$

According to formula (1), the vector angle is calculated, and the angle between the proportion vector of the three industries and the corresponding coordinate axis is used to reflect and define the advanced value of industrial structure. The specific formula is as follows:

$$sadvance_{i,j} = \sum_{i=1}^{3}\sum_{j=1}^{i}\alpha_j \tag{2}$$

$sadvance_{i,t}$ The higher the value of $sadvance_{i,t}$ is, the higher the level of marine fishery industrial structure is.

The rationalization of marine fishery industrial structure (*strationlize*). It is measured by structural entropy based on previous research [44]. The calculation formula of marine fishery industrial structure entropy index is as follows:

$$strationlize_{i,t} = \sum_{j=1}^{n}\left[ X_{i,j,t} ln\left( \frac{1}{X_{i,j,t}} \right) \right] \tag{3}$$

In formula (3), *strationlize*$_{i,t}$ indicates the entropy index of marine fishery industry structure of area *i* in period *t*, and $X_{i,j,t}$ indicates the proportion of industry *j* in total output value of marine fishery in period *t* in area *i*, and the type of marine fishery industry is represented by *j*, $j \in N^{+}$. According to formula (3), the entropy index of marine fishery industry structure reaches the maximum value when the proportion of marine fishery industries in the total output value of marine fishery is equal. If the development of various industries in marine fisheries is uneven, the entropy index of industrial structure will be smaller. On the contrary, the higher the entropy index of marine fishery industrial structure is, the more balanced and coordinated the development of various industries will be, and the more reasonable the industrial structure will be.

The softening of the marine fishery industry structure(*ssoften*). The proportion of marine fishery tertiary industry in marine fishery GDP is used as the measurement index to measure the softening of marine fishery industrial structure (*ssoften*). The calculation formula is as follows:

$$ssoften_{i,t} = \frac{ftiv_{i,t}}{fpv_{i,t}} \qquad (4)$$

In formula (4), *ssoften*$_{i,t}$ represents the softening level of marine fishery industrial structure in region *i* in time *t*; *ftiv*$_{i,t}$ represents the output value of marine fishery tertiary industry (marine fishery circulation and service industry) in time *t* of area *i*; and *fpv*$_{i,t}$ represents the gross output value of marine fishery in time *t* of area *i*. The higher the value of *ssoften*$_{i,t}$ is, the higher the softening level of marine fishery industry structure is, and the service level of marine fishery is significantly improved.

The aquaculture-catching structure of Marine Fisheries (*mfcs*). It is mainly used to reflect the changes of the supply structure of fresh seafood in the development of marine fisheries. It is measured by the proportion of mariculture output and marine fishing yield. The calculation formula is as follows:

$$mfcs_{i,t} = \frac{fqi_{i,c,t}}{fqi_{i,f,t}} \qquad (5)$$

In formula (5), *mfcs*$_{i,t}$ is the coefficient of marine fishery aquaculture-catching structure in area *i* in the year *t*, *fqi*$_{i,c,t}$ represents the marine fishery aquaculture output in area *i* in the year *t*, and *fqi*$_{i,f,t}$ represents the marine fishery catching production in area *i* in the year *t*.

Marine fishery processing coefficient (*mfsp*). It is measured by the proportion of the number of processed seawater products to the number of fresh seafood. The calculation formula is as follows:

$$mfsp_{i,t} = \frac{fpq_{i,t}}{tfq_{i,t}} \qquad (6)$$

In formula (6), *mfsp*$_{i,t}$ is the processing coefficient of marine products in the year *t* of area *i*, *fpq*$_{i,t}$ represents the quantity of sea water processing products of marine fishery in area *i* in the year *t*, and *tfq*$_{i,t}$ represents the quantity of fresh seafood of marine fishery in area *i* in the year *t*. Since the quantity of sea water processed products is less than the quantity of seafood used for product processing, and the quantity of seafood used for processing is less than the total amount of fresh seafood, the value of *fpq*$_{i,t}$ should belong to [0, 1].

## Model building

This paper establishes a panel regression model with the industrial structure of marine fishery as the core explanatory variable and the total factor productivity of marine fishery as the explanatory variable. Because the factors affecting TFP of marine fishery are not only the industrial structure, it is necessary to introduce control variables to improve the measurement accuracy. At present, there is no consistent standard for the selection of control variables, which makes the selection of control variables more random. In order to overcome the randomness of selecting control variables, using the method of Gan and Zheng [30], the interaction between marine fishery industrial structure and TFP is used as the control variable to replace all factors affecting TFP of marine fishery except marine fishery industrial structure. In order to avoid collinearity and heteroscedasticity after introducing control variables, the corresponding explanatory variables are logarithmically treated. The regression model was obtained.

$$mftfp_{i,t} = \alpha_0 + \alpha_1 mftfp_{i,t-1} + \beta_j \ln M \, FS_{i,j,t} + \eta_j mftfp_{i,t} \ln M \, FS_{i,j,t} + \varepsilon_{i,t} + \mu_i \qquad (7)$$

In the model (7), $i$ represents the region; $t$ represents the time; $\beta_j = [\beta_1, \beta_2, \beta_3, \beta_4, \beta_5]$ represents the coefficient set of the influence of different industrial structure changes of marine fishery on its total factor productivity; $\eta_j = [\eta_1, \eta_2, \eta_3, \eta_4, \eta_5]$ represents the set of influence coefficients of control variables; $MFS_{i,j,t} =$

$[sadvance_{i,1,t}, \; strationlize_{i,2,t}, ssoften_{i,3,t}, \; mfcs_{i,4,t}, \; mfsp_{i,5,t}]^T$ represents the set of different measurement indicators of marine fishery industrial structure in the year $t$ of area $i$; $mfftp_{i,t}$ represents the total factor productivity of marine fishery in area $i$ in the year $t$, $\alpha_0$ is a constant; $\alpha_1$ represents the influence coefficient of the total factor productivity of marine industry with one lag period on the total factor productivity of marine fishery in the current period; $u_i$ and $\varepsilon_{it}$ represent the intercept term of individual heterogeneity and the disturbance term changing with individual and time, respectively.

Model (7) is processed by difference to eliminate the influence of individual effect on model estimation. The difference model to be estimated is obtained:

$$\Delta mftfp_{i,t} = \alpha_0 + \alpha_1 \Delta mftfp_{i,t-1} + \beta_j \Delta \ln M \, FS_{i,t} + \eta_j (\Delta mftfp_{i,t} * \Delta \ln M \, FS_{i,t}) + \Delta \varepsilon_{i,t} \qquad (8)$$

In order to avoid the pseudo regression caused by the correlation of random disturbance terms, the standard deviation of panel robustness is introduced to eliminate the influence. In addition, due to the introduction of interaction items with explained variables in the model, it may lead to endogenous problems in the model, so it is necessary to introduce tool variables to eliminate them. In order to avoid the randomness and deviation in the selection of control variables, the lag and difference items of explanatory variables are taken as tool variables, and dynamic panel generalized moment meter (GMM) is selected for simulation. In order to improve the effectiveness of tool variables and avoid over identification of tool variables, Hansen and Sargan statistical methods are used to test the over identification.

## Data source and description

The study areas mainly include Tianjin, Hebei, Liaoning, Jiangsu, Zhejiang, Fujian, Shandong, Guangdong, Guangxi and Hainan. Due to the incomplete data of marine fishery development in Shanghai, considering the consistency of the data, it was not included in the study area. The sample data are from China fishery statistical yearbook, China Agricultural Statistics Yearbook and China Statistical Yearbook, mainly intercepting the marine fishery economic development data from 2003 to 2016.

As the data of sea water breeding, seafood processing, marine fishing machine repair, marine fishing rope net manufacturing, marine fishery feed and medicine, marine fishery circulation, marine fishery (storage) transportation, marine leisure fishery and other data are not listed separately in the relevant statistical yearbook, this paper adjusts the relevant data by using the adjustment method of marine fishery economic data by Wang Bo et al. [42]. At the same time, in order to improve the accuracy of the empirical test, taking 2002 as the base period, the price index of fishery GDP was used to adjust the output data to eliminate the impact of inflation rate.

## Experimental testing and results

**Stability test.** Fisher's test of different root unit root and the same root unit root were used to test. The test results (Table 2) show that the variables *mftfp*, *lnsadvance*, *lnsrationalize*, *lnssoften*, *lnmfcs*, *lnmfsp*, *mftfp·lnMFS* have passed the 5% significance test, indicating that there is no unit root in the panel data, and the data is stable, so the model estimation can be carried out.

## Regression results

Before the model estimation, Hansen test is carried out. The results show that the random effect estimation is better than the fixed effect estimation at the significance level of 5%. Therefore, we choose the random effect to estimate the static model (model 1), at the same time, the dynamic impact of the change of marine fishery industrial structure on the total factor productivity of marine fishery is analyzed by using the differential GMM method (model 2). The specific regression results are shown in Table 3.

The results in Table 3 show that $R^2$ in model 1 is 0.999, which indicates that the regression model fits the observed values well, and the *F* statistic has passed the significance level test of 1%, indicating that the linear relationship between the explained variables and all the explanatory variables in the model is significant on the whole, and the overall fitting effect is good. Among the explanatory variables, except that the rationalization of marine fishery industrial structure did not pass the *t* test, the other explanatory variables passed the significance level test of 5%. In model 2, Wald chi2 passed the significance test of 1%, indicating that the overall regression effect of the model is relatively good. At the same time, the probability values

**Table 2. Test results of variable stability.**

| variables | Fisher-ADF test | | | | LLC test | Test results |
|---|---|---|---|---|---|---|
| | **P** | **Z** | **L***$^{*}$ | **P$_m$** | **Adjusted t***$^{*}$ | |
| *mftfp* | 95.294*** | -7.348*** | -8.346*** | 11.905*** | -6.523*** | stable |
| *lnsadvance* | 57.784*** | -4.176*** | -4.479*** | 5.974*** | -3.942*** | stable |
| *lnsrationalize* | 32.384** | -1.691** | -1.728** | 1.958*** | -1.871** | stable |
| *lnssoften* | 61.168*** | -4.950*** | -5.135*** | 6.509** | -1.817** | stable |
| *lnmfcs* | 34.271** | -2.560*** | -2.474*** | 2.257*** | -3.789*** | stable |
| *lnmfsp* | 36.980** | -2.643*** | -2.629*** | 2.685*** | -3.298*** | stable |
| *lnsadvance*mftfp* | 95.563*** | -7.376*** | -8.372*** | 11.948*** | -6.734*** | stable |
| *lnsrationalize*mftfp* | 43.051*** | -2.806*** | -2.929*** | 3.645*** | -2.507*** | stable |
| *lnssoften*mftfp* | 62.482*** | -5.141*** | -5.339*** | 6.717*** | -2.734*** | stable |
| *lnmfcs*mftfp* | 48.355*** | -4.080*** | -4.007*** | 4.483*** | -3.553*** | stable |
| *lnmfsp*mftfp* | 63.366*** | -4.985*** | -5.349*** | 6.857*** | -5.752*** | stable |

Note: * *, * *, denote significant levels of 5% and 1% respectively. P is the inverse chi square transformation; Z is the inverse normal transformation; L* is the inverse logic transformation; Pm is the modified inverse chi square transformation.

**Table 3. Regression results of the impact of changes in marine fishery industrial structure on its total factor productivity.**

| Explanatory variables | model 1 | model 2 | Explanatory variables | model 1 | model 2 |
|---|---|---|---|---|---|
| $mftfp_{i,t-1}$ | — | -0.002 (-1.19) | $lnmfcs_{i,t}*mftfp_{i,t}$ | 0.009*** (7.47) | 0.010** (2.09) |
| $lnsadvance_{i,t}$ | -0.564*** (-86.65) | -0.563*** (-12.75) | $lnmfsp_{i,t}*mftfp_{i,t}$ | -0.010*** (-12.41) | -0.008*** (-7.63) |
| $lnsrationalize_{i,t}$ | -0.003 (-0.70) | 0.023** (2.32) | _cons | 1.010*** (87.67) | — |
| $lnssoften_{i,t}$ | 0.032*** (25.16) | 0.025** (7.84) | $R^2$ | 0.999 | — |
| $lnmfcs_{i,t}$ | -0.009*** (-7.09) | -0.012* (-1.80) | AR(2) | — | 0.466 |
| $lnmfsp_{i,t}$ | 0.010*** (11.03) | 0.012*** (5.42) | Sargan test (p-value) | — | 0.745 |
| $lnsadvance_{i,t}*mftfp_{i,t}$ | 0.561** (196.31) | 0.572*** (67.25) | Hansen test (p-value) | — | 1.000 |
| $lnsrationalize_{i,t}*mftfp_{i,t}$ | -0.001*** (-0.22) | -0.024*** (-2.86) | F / Wald chi2 | 1.76e+06*** | 72511.0*** |
| $lnssoften_{i,t}*mftfp_{i,t}$ | -0.031*** (-28.86) | -0.029*** (-5.82) | Number of obs | 120 | 120 |

Note: *, * *, * *, denote the significant level of 10%, 5% and 1% respectively, and the value in () indicates the value of Z. Model 1 is the result of random effect model; model 2 is the GMM Estimation of the explanatory variables which lag 2–3 order as tool variables.

corresponding to AR (2), Sargan's test results and Hansen's test results are greater than 5%, indicating that the tool variables selected are reasonable and there is no over recognition. The regression results of model 2 show that the TFP of marine fishery fails to pass the significance level test of 10%, the fishery structure of marine fishery passes the significance test of 10%, and other explanatory variables pass the significance test of 5%. The regression results of model 1 and model 2 show that only the effect of rationalization of marine fishery industrial structure on TFP of marine fishery is different, while the effect of other explanatory variables on TFP of marine fishery is basically the same. Considering the dynamic development of marine fishery economy, we adopt the regression results of dynamic panel model.

## Discussion

### Analysis of the effect of "structural dividend"

(1) The rationalization of marine fishery industrial structure has significant "structural dividend" on the improvement of total factor productivity of marine fishery. However, the interaction between the rationalization of marine fishery industrial structure and the total factor productivity of marine fishery shows that the interaction with other economic factors will inhibit the improvement of TFP of marine fishery. At the same time $|\beta_2/\eta_2|<2$, this shows that when the rationalization value of marine fishery industrial structure is small, the unreasonable industrial structure will not inhibit the improvement of total factor productivity of marine fishery. The rationalization of marine fishery industrial structure can avoid the low efficiency caused by excessive concentration of marine fishery resources and elements through the reallocation of production factors or resources, and reduce the lack of kinetic energy caused by the shortage of resource elements in marine fishery development, so as to realize the dynamic balance between resource allocation and industrial development, improve the allocation efficiency of production factors, and promote the coordination of marine fishery industries Adjust development and improve the efficiency of marine fishery development. Therefore, the rationalization of marine fishery industrial structure has a significant "structural dividend" to improve the total factor productivity of marine fishery. This is consistent with conclusion of Han et al. [45] on the impact of industrial structure rationalization on ecological effects.

(2) The softening of marine fishery industrial structure also has "structural dividend" on the improvement of marine fishery quality. However, from the interaction between the softening of marine fishery industrial structure and the total factor productivity of marine fishery, its

effect on the production factors of marine fishery is negative, which indicates that the interaction with other influencing factors will inhibit the improvement of TFP of marine fishery. At the same time $|\beta_3/\eta_3|<1$, it shows that the softening of industrial structure will not inhibit the improvement of total factor productivity of marine fisheries when the softening value of industrial structure is small. The improvement of the softening degree of marine fishery industry structure will promote the development of marine fishery science and technology, information, education and other related industries, and improve the comprehensive ability and level of marine fishery science and technology innovation. The innovative application of new fishery technology will gradually expand the technology spillover effect of marine fishery. Advanced fishery equipment and high-level aquaculture technology will improve the operation capacity and production efficiency of the three marine fishery industries. Technology innovation and transformation are easily affected by other factors (such as scientific and technological personnel, research and development costs, transformation mechanism, etc.), reducing the efficiency of science and technology transformation will affect the promotion of marine fishery science and technology to the development of marine fishery to a certain extent, but this impact will not inhibit the impact of the softening of marine fishery industrial structure on promoting the total factor productivity of marine fisheries.

(3) The improvement of the processing coefficient of marine fishery is conducive to the improvement of its total factor productivity, which has the effect of "structural dividend". This is consistent with Zhou et al 's research conclusion [46]. The internal reason for this result is that the marine fishery processing industry has higher requirements for marine fishery processing technology, and the realization of diversified processing of marine products depends on the fishery technology innovation. The improvement of marine fishery processing technology will not only improve the processing capacity of marine fishery, but also improve the production efficiency of marine fishery [47]. Therefore, the improvement of marine fishery processing coefficient has a significant structural dividend on TFP of marine fishery. However, from the interaction between marine fishery processing coefficient and total factor productivity of marine fishery, its effect on marine fishery production factors is negative, indicating that the interaction with other influencing factors will inhibit the improvement of total factor productivity of marine fishery. At the same time $|\beta_2/\eta_2|>1$, this shows that the positive effect of marine fishery processing coefficient is higher than the negative effect of interaction term with other influencing factors, indicating that the "structural dividend" of marine fishery processing coefficient on TFP of marine fishery will not be reduced by other factors.

## Analysis of the effect of "structural negative benefit"

(1) The advancement of marine fishery industry structure will inhibit the improvement of TFP of marine fishery, which fails to meet the "structural dividend hypothesis". However, from the perspective of the interaction between the advancement of marine fishery industrial structure and the total factor productivity of marine fishery, its effect on marine fishery production factors is positive, which indicates that the interaction with other economic factors will improve the TFP of marine fishery. At the same time $|\beta_1/\eta_1|<1$, this shows that unless the marine fishery industrial structure can be rapidly advanced, otherwise simply pursuing the marine fishery industrial structure advancement will reduce the improvement of the total factor productivity of marine fishery to a certain extent.

Total factor productivity (TFP) of marine fishery is the part of the productivity in which capital and labor are deducted. At present, the advancement of marine fishery industrial structure mainly promotes the transfer of marine fishery production factors from the primary industry to the secondary and tertiary industries, and the transfer of capital and labor

resources from low productivity sectors to high productivity sectors. In the short term, the marine fishery economic growth caused by the advancement of industrial structure still depends on the expansion of factor scale and the improvement of utilization efficiency after transfer, However, the contribution of technological progress and technical efficiency is relatively low, so it has a negative effect on improving the total factor productivity of marine fisheries. This coincides with Gan et al 's view [48] that the advancement of the industrial structure in some regions is not conducive to economic development. Although the advancement of marine fishery industrial structure has a "structural negative benefit" effect on its total factor productivity, when the industrial structure is upgraded to a certain height, it will promote the improvement of TFP through other factors (such as technology upgrading, management mode improvement, product quality improvement, etc.).

(2) The effect of the aquaculture-catching structure of marine fishery on the TFP of marine fishery is significantly negative, which indicates that the change of fishery structure will restrain the increase of TFP of marine fishery to a certain extent. However, from the interaction between the structure of marine fishery and TFP of marine fishery, its effect on the production factors of marine fishery is positive, which indicates that it has an impact on other economic factors. The interaction of elements will improve the total factor productivity of marine fisheries. At the same time $|\beta_4/\eta_4|>1$, this shows that the negative effect of the structure of marine fishery on the improvement of TFP is small, and it can promote the TFP of marine fishery through the interaction with other influencing factors.

This result is determined by the stage characteristics of marine fishery development. At present, mariculture is the main mode of seafood supply, which is still driven by labor and capital. In the low level of fishery technology, the development of mariculture industry is mainly based on the expansion of capital and labor production factors, especially in the purchase of aquaculture equipment (such as cage, fishing boat, raft, hanging cage, etc.), renting aquaculture sea area, and employing a large number of labor force engaged in simple processing of marine products. Although the advancement of technological progress has improved, it is not obvious [49], and the momentum for increasing the total factor productivity of the marine fishery economy is insufficient.

## Conclusion and suggestions

We design a model of the influence of the change of marine fishery industrial structure on the total factor productivity of marine fishery from the perspective of fishery supply side, and makes an empirical test by using the development data of marine fishery in ten coastal areas. The results show that the change of marine fishery industry structure has a structural effect on the total factor productivity of marine fishery, but the structural effect under different dimensions is different. Among them, the effect of rationalization, softening and processing coefficient of marine fishery industrial structure on total factor productivity of marine fishery has "structural dividend" effect, that is, the rationalization, softening and improvement of deep processing degree of marine fishery industrial structure are conducive to improving the total factor productivity of marine fishery; while the advancement and aquaculture-catching structure of marine fishery have a "structural negative" effect on TFP, but the interaction with other influencing factors could promote the improvement of TFP of marine fishery. This shows that it can't promote the improvement of TFP by relying solely on the advancement of the marine fishery industry structure and the adjustment of the aquaculture-catching structure. In the process of advancement and adjustment of the aquaculture-catching structure, we should strengthen the coordinated advancement with other production factors. Therefore, we put forward some suggestions on the improvement of total factor productivity of marine fishery.

(1) Promoting the advancement and rationalization of the marine fishery industrial structure, and enhancing the "structural dividend" effect. The promotion of marine fishery value chain depends on the evolution of industrial structure, but the evolution of industrial structure will lead to economic fluctuations, which will reduce the efficiency of industrial development to a certain extent. In order to reduce the adverse effects caused by the evolution of advanced industries, on the one hand, in the process of industrial advancement, we should pay more attention to the rationalization of industrial structure, optimize the allocation structure of marine fishery resources, avoid reducing the efficiency of economic development due to excessive concentration of resources in a certain field, and reasonably guide and arrange the allocation of talents, funds and other high-quality marine fishery resources in various fields of marine fisheries [50]. In the high-end development mode, the balanced development of marine fishery industry should be carried out. At the same time, the advancement rate of marine fishery industrial structure should be improved to reduce the lasting impact caused by slow evolution.

(2) Promoting the key adjustment of marine fishery science and technology innovation and achievements transformation, to accelerate the transfer of marine fishery technology to the secondary industry and improve the progress of marine fishery technology [51]. We should accelerate the cross-border integration of Marine Fisheries in tackling key scientific and technological problems, gradually expand the scope of marine fishery science and technology services, reverse the service concept that marine fishery science and technology only provides support for marine fishing and mariculture, and transfer to the marine secondary industry with high added value. Focusing on the development needs of offshore, ecological, cage, factory farming and other aquaculture modes, we should strengthen technical cooperation and exchange with Norway, Japan, the United States and other countries, develop and manufacture high-end green equipment for marine fishery, improve the technical level and R & D capacity of fishery machinery, and enhance the supply capacity of marine fishery equipment. Comply with the concept of marine fishery ecological development, create low pollution and high standard marine fishery buildings, and provide high-quality artificial reefs for the construction of marine ranches [52]. We will promote the construction of mechanization, intelligence and ecology of marine fishery, and promote the progress of fishery technology and the improvement of industrial efficiency.

(3) Promoting the development of marine products deep processing industry through cross-border integration. To get rid of the current development mode of simple processing of marine fishery, combined with the development needs and new trends of medical and pharmaceutical industry, health care industry, beauty industry and chemical industry, relying on the innovation of marine fishery processing technology, extract the effective active substances of marine organisms, especially in medical and health products (such as extracting polysaccharides and taurine from shellfish, producing anti-tumor, anti-aging and improving immunity). We will promote the deep processing of marine products, extend the industrial chain with marine fishery innovation chain, and enhance the value chain with the industrial chain. At the same time, we should pay attention to the comprehensive utilization of marine products and the comprehensive utilization of marine products.

## Supporting information

**S1 Data.**
(XLS)

## Author Contributions

**Data curation:** Hongzhi Zhang.

**Formal analysis:** Bo Wang.

**Methodology:** Bo Wang.

**Supervision:** Limin Han.

**Writing – original draft:** Bo Wang.

**Writing – review & editing:** Bo Wang.

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
