## [Decision Letter · Decision Letter 0]

6 Aug 2021

PONE-D-21-20148

Analysis on the structure effect of marine fishery total factor productivity under high-quality development in China

PLOS ONE

Dear Dr. Hongzhi Zhang,

Thank you for submitting your manuscript to PLOS ONE. After careful consideration, we feel that it has merit but does not fully meet PLOS ONE’s publication criteria as it currently stands. Therefore, we invite you to submit a revised version of the manuscript that addresses the points raised during the review process.

We look forward to receiving your revised manuscript.

Kind regards,

Carlos Alberto Zúniga-González, Ph.D

Academic Editor

PLOS ONE

“Project Support: Youth Project of Natural Science Foundation of Guangxi Science and Technology Department (Number: 2018GXNSFBA050010); Youth Project of National Social Science Foundation of China ( Number: 19CJY023); Guangxi Key R & D Plan ( Number: Guike AB1850023)” 

4. Please ensure that you refer the all Figure (1-7) in your text as, if accepted, production will need this reference to link the reader to the figure.

Additional Editor Comments (if provided):

Dear authors, in order to improve the quality of your manuscript, we especially suggest that you include citations in the discussion and land a little on the conclusions according to the objective pursued in the research. In the same way, I suggest that you revise the writing to clarify the observations of the reviewer number two, in the same way I suggest the following references

1) González, C. A. Z. (2011). Technical efficiency of organic fertilizer in small farms of Nicaragua: 1998-2005. African Journal of Business Management, 5(3), 967-973. Available from publons.com/p/11272633/

2) Dios-Palomares, R. (2015). 7. Analysis of the Efficiency of Farming Systems in Latin America and the Caribbean Considering Environmental Issues. Revista Cientifica-Facultad de Ciencias Veterinarias, 25(1). Available in publons.com/p/3106827/

3) Zuniga González, C. (2020). Total factor productivity growth in agriculture: Malmquist index analysis of 14 countries, 1979-2008. REICE: Revista Electrónica De Investigación En Ciencias Económicas, 8(16), 68-97. https://doi.org/10.5377/reice.v8i16.10661 Available from publons.com/p/36247914/

4) Blanco-Orozco, N., Arce-Díaz, E., & Zúñiga-Gonzáles, C. (2015). Integral assessment (financial, economic, social, environmental and productivity) of using bagasse and fossil fuels in power generation in Nicaragua. Revista Tecnología en Marcha, 28(4), 94-107. Available from publons.com/p/32281799/

Reviewers' comments:

Reviewer's Responses to Questions

**Comments to the Author**

1. Is the manuscript technically sound, and do the data support the conclusions?

Reviewer #1: Yes

Reviewer #2: Partly

2. Has the statistical analysis been performed appropriately and rigorously? 

Reviewer #1: Yes

Reviewer #2: I Don't Know

3. Have the authors made all data underlying the findings in their manuscript fully available?

Reviewer #1: Yes

Reviewer #2: Yes

4. Is the manuscript presented in an intelligible fashion and written in standard English?

Reviewer #1: Yes

Reviewer #2: No

5. Review Comments to the Author

Reviewer #1: The article has been presented with technical professionalism and the data of the DEAP model have been provided in an appropriate way. The statistical treatment of the variables of the study model was appropriate.Therefore, from the scientific point of view it seems to me that the article presents the technical qualities for its publication.

Reviewer #2: 1. the abstract is not well written, and it needs an conclusion and sussestion;

2. in the Question raising and literature review (part 1), there are no references for the datas;

3. there are two many simple figures in the manuscript, and many freshwater data is needless, in the figure 4, freshwater aquaculture has been written to mreshwater aquaculture;

4. i can not found any references in the discussion and Conclusion and suggestions, please added it

6. PLOS authors have the option to publish the peer review history of their article (what does this mean?). If published, this will include your full peer review and any attached files.

Reviewer #1: **Yes: **Napoleon Vicente Blanco Orozco

Reviewer #2: No

---

## [Author Response · Author response to Decision Letter 0]

11 Oct 2021

Dear editors

Thank you very much for the editors and review experts for their suggestions for this paper. First, I reply to the amendment suggestions one by one. The content is as follows:

1.Please ensure that your manuscript meets PLOS ONE's style requirements, including those for file naming. 

Reply: The paper has been revised according to the format requirements.

2.Thank you for stating the following financial disclosure:

“Project Support: Youth Project of Natural Science Foundation of Guangxi Science and Technology Department (Number: 2018GXNSFBA050010); Youth Project of National Social Science Foundation of China ( Number: 19CJY023); Guangxi Key R & D Plan ( Number: Guike AB1850023)” 

Reply: 

Youth Project of Natural Science Foundation of Guangxi Science and Technology Department (Number: 2018GXNSFBA050010) and Youth Project of National Social Science Foundation of China ( Number: 19CJY023) provide support in study design and analysis.The paper is one of the phased research results of this project(Youth Project of Natural Science Foundation of Guangxi Science and Technology Department (Number: 2018GXNSFBA050010). 

Guangxi Key R & D Plan ( Number: Guike AB1850023)provided basic materials for writing the thesis (such as marine fishery development data).

3. In your Data Availability statement, you have not specified where the minimal data set underlying the results described in your manuscript can be found. PLOS defines a study's minimal data set as the underlying data used to reach the conclusions drawn in the manuscript and any additional data required to replicate the reported study findings in their entirety. All PLOS journals require that the minimal data set be made fully available.

Reply: For the data involved in the paper, the author has sent it to the journal in the form of a file (excel)

4.Please ensure that you refer the all Figure (1-7) in your text as, if accepted, production will need this reference to link the reader to the figure.

Reply: It has been verified, and the name of the relevant figure is marked in the text.

5.In order to improve the quality of your manuscript, we especially suggest that you include citations in the discussion and land a little on the conclusions according to the objective pursued in the research. 

Reply: Adding this reference: 

Dios-Palomares, R. (2015). Analysis of the Efficiency of Farming Systems in Latin America and the Caribbean Considering Environmental Issues. Revista Cientifica-Facultad de Ciencias Veterinarias, 25(1). Available in publons.com/p/3106827/

Reviewer #2: 

1.the abstract is not well written, and it needs an conclusion and sussestion;

Reply:The abstract has been revised and refined. We newly added essay viewpoint. as follows:

 “We believe that simply pursuing the advanced evolution of the industrial structure is not conducive to sustainable development of fisherie. Under the pursuit of the rationalization of the marine fishery industry structure, by promoting the coordinated evolution of marine fisheries advancement, aquaculture-catching structure and other factors, the "structural dividend" effect can be enhanced and the fishery can achieve sustainable development.”

2.in the Question raising and literature review (part 1), there are no references for the datas;

Reply: Added data sources in the form of footnotes,as follows:

 Data source: "China Fishery Statistical Yearbook". If there is no special description in the following text, the data source is the same.

3.there are two many simple figures in the manuscript, and many freshwater data is needless, in the figure 4, freshwater aquaculture has been written to mreshwater aquaculture;

Reply: The freshwater data has been deleted, and this figure has been updated. as follows.

4.I can not found any references in the discussion and Conclusion and suggestions, please added it.

Reply: Based on the analysis and conclusions of the research results of the thesis, relevant references have been added

Han YH, Zhang F, Huang LX, Peng KM, Wang XB. Does industrial upgrading promote eco-efficiency? ─A panel space estimation based on Chinese evidence. Energy Policy. 2021; 154: 112286.doi: org/10.1016/j.enpol.2021.112286.

Zhou Y, Kong Y, Sha J, Wang HK. The role of industrial structure upgrades in eco-efficiency evolution: Spatial correlation and spillover effects. Sci Total Environ. 2019; 687: 1327-1336.doi: 10.1016/j.scitotenv.2019.06.182.

Xiang XM. The transformation and upgrading path of my country's marine fishery from the perspective of agricultural supply-side structural reform. Social Sciences in Guangdong. 2017; (05):23-29.

Gan C, Zheng RG, Yu DF. An empirical study on the effects of industrial structure on economic growth and fluctuations in China. Economic Research Journal. 2011; 46(05):4-16,31. doi: 10.1093/hmg/ddr440.

Zhang Y, J Ji. The decoupling and influencing factors analysis of blue granary Eco-Economy system. Journal of Agrotechnical Economics. 2020; (04):94-106. doi: 10.13246/j.cnki.jae.2020.04.006.

Yang L, Su X. Research on the objectives and implementation paths of the optimization and upgrading of the marine fishery industrial structure from the perspective of industrial ecology. Issues in Agricultural Economy. 2010; 31(10): 99-105.doi: 10.13246/j.cnki.iae.2010.10.011.

Shao QL, Chen LJ, Zhong RY, Weng HT. Marine economic growth, technological innovation, and industrial upgrading: A vector error correction model for China. Ocean & Coastal Management. 2021; 200: 105481.doi: 10.1016/j.ocecoaman.2020.105481.

Lin XH, Zheng L, Li WW. Measurement of the contributions of science and technology to the marine fisheries industry in the coastal regions of China. Marine Policy. 2019; 108: 103647.doi: 10.1016/j.marpol.2019.103647.

---

## [Decision Letter · Decision Letter 1]

28 Oct 2021

Analysis on the structure effect of marine fishery total factor productivity under high-quality development in China

PONE-D-21-20148R1

Dear Dr. Hongzhi Zhang,

We’re pleased to inform you that your manuscript has been judged scientifically suitable for publication and will be formally accepted for publication once it meets all outstanding technical requirements.

Kind regards,

Carlos Alberto Zúniga-González, Ph.D

Academic Editor

PLOS ONE

Additional Editor Comments (optional):

Congratulations on the effort to improve the quality of your manuscript.

Reviewers' comments:

Reviewer's Responses to Questions

**Comments to the Author**

1. If the authors have adequately addressed your comments raised in a previous round of review and you feel that this manuscript is now acceptable for publication, you may indicate that here to bypass the “Comments to the Author” section, enter your conflict of interest statement in the “Confidential to Editor” section, and submit your "Accept" recommendation.

Reviewer #2: All comments have been addressed

2. Is the manuscript technically sound, and do the data support the conclusions?

Reviewer #2: Yes

3. Has the statistical analysis been performed appropriately and rigorously? 

Reviewer #2: Yes

4. Have the authors made all data underlying the findings in their manuscript fully available?

Reviewer #2: Yes

5. Is the manuscript presented in an intelligible fashion and written in standard English?

Reviewer #2: Yes

6. Review Comments to the Author

Reviewer #2: The authors have done a good job of addressing my previous concerns and I believe the manuscript is very close to publishable

7. PLOS authors have the option to publish the peer review history of their article (what does this mean?). If published, this will include your full peer review and any attached files.

Reviewer #2: No

---

## [Editor Report · Acceptance letter]

2 Nov 2021

PONE-D-21-20148R1 

Analysis on the structure effect of marine fishery total factor productivity under high-quality development in China 

Dear Dr. Zhang:

I'm pleased to inform you that your manuscript has been deemed suitable for publication in PLOS ONE. Congratulations! Your manuscript is now with our production department. 

Kind regards, 

on behalf of

Dr. Prof. Carlos Alberto Zúniga-González 

Academic Editor

PLOS ONE